# Robotic Single-Site Hysterectomy in Gynecologic Benign Pathology: A Systematic Review of the Literature

**DOI:** 10.3390/medicina59020411

**Published:** 2023-02-20

**Authors:** Gaetano Riemma, Francesca Pasanisi, Antonella Reino, Maria Cristina Solazzo, Carlo Ronsini

**Affiliations:** Department of Woman, Child and General and Specialized Surgery, University of Campania “Luigi Vanvitelli”, 80138 Naples, Italy

**Keywords:** robotic single site, hysterectomy, benign pathology, outcomes, safety, feasibility

## Abstract

*Background and objectives:* Total hysterectomy is one of the most common gynecologic surgical procedures and it is mainly performed for benign pathologies. The introduction of robotic single-site surgery (RSS) as an acceptable alternative to laparoendoscopic surgery combines the advantages of robotics with the aesthetic result of a single incision. This study aims to review the existing literature on a single-site robotic hysterectomy in patients with benign pathologies and verify its safety and feasibility. *Materials and Methods*: Following the recommendations in the preferred reporting items for systematic reviews and meta-analyses (PRISMA) statement, FP and AR systematically screened the PubMed, Embase, and Scopus databases. No temporal or geographical limitation was discriminatory. Studies containing data about feasibility and safety were included. *Results*: From 219, only eight studies met the inclusion criteria, and a total of 212 patients were included with a mean patient age of 45.42 years old (range 28–49.5 years old) and a mean BMI of 25.74 kg/m^2^ (range 22–28.5 kg/m^2^). The mean presurgical time, including port placement and docking time, was 15.56 (range 3–30) minutes. Mean console time was reported in six studies and is 83.21 min (range 25–180 min). The mean operative time is 136.6 min (range 60–294 min) and the mean blood loss is 43.68 mL (range 15–300 mL). Only two patients in the total analyzed had intraoperative complications and no conversion to LPT occurred. The median hospital stay was 1.71 days (range 0.96–3.5 days). The postoperative complication rate was estimated at 1.4% (vaginal bleeding). *Conclusions:* Our review supports the safety and feasibility of robotic single-site hysterectomy for benign gynecological diseases.

## 1. Introduction

Hysterectomy is the most commonly performed surgical procedure in gynecological practice and, in 70% of cases, it represents the preferred treatment for benign pathologies and conditions, especially in cases of uterine bleeding, symptomatic fibroids, uterus prolapse, and adenomyosis [1,2].

Most hysterectomies in the United States are performed via laparotomic approach, through a median or transverse incision of the abdominal wall. Nevertheless, different studies showed how minimally invasive surgery (MIS) for benign gynecologic conditions has several benefits compared to open surgery [3]. For instance, MIS takes advantage of the use of one up to three port sites to access the abdominal cavity. Surgical instruments and a camera are introduced through the trocars inside the abdomen. Therefore, laparoscopy provides direct, panoramic, and magnified visualization of the pelvis, improving the exposure of anatomical structures. Moreover, thanks to its lower invasiveness, MIS seems to be associated with a decrease in intra and postoperative complications and bleeding, a reduction in wound complications and pain, and a shorter hospital stay and recovery [4,5].

Aesthetic results are also an important factor, particularly in young patients undergoing surgical procedures [6]. On this line, a new method, laparoendoscopic surgery in a single site (LESS), has been introduced in minimally invasive surgery. It uses a single port incision through the umbilical scar to access the abdomen. Moreover, reducing the number of abdominal incisions is associated with a lower risk of port related morbidities, such as scar infections, hematomas, and nerve and vessel lesions [6,7,8]. However, LESS has many limitations, such as loss of door triangulation and instrumental collisions [7]. To overcome these challenges, robotic single-site surgery has been approved as an alternative to LESS. Robotic Single Site (RSS) has the advantage of aesthetic results combined with a system of wristed instruments, allowing wider and more accurate movements, tremor filtration, improved ergonomics, and a three-dimensional view [9,10]. The use of the da Vinci surgical system also allowed surgeons not adept at standard laparoscopy to approach minimally invasive procedures [11]. Initial studies have shown that this technique is also safe and effective and can help to solve the technical limitations found in LESS [9].

The purpose of our review is to analyze data about the feasibility and safety of RSS hysterectomy for benign causes in the existing literature thus far.

## 2. Materials and Methods

Records selection observed the PRISMA (preferred reporting items for systematic reviews and meta-analysis) guidelines [12].

### 2.1. Search Methods

A systematic search from PubMed, Embase, and Scopus databases was performed in October 2022. Only papers published in the last 9 years were included. This frame time represents the entire period since FDA approval of the RSS device. No restriction of the country was performed. The string of idioms adopted to identify studies fitting to the topic of the review was: “Hysterectomy” AND “robotic single site” NOT (“neoplasm” OR “cancer”).

### 2.2. Studies Selection

F.P. and A.R. independently carried out studies selection through systematic research from three databases: PubMed, Scopus, and Embase. C.R resolved the discrepancies during the data collection.

The inclusion criteria were: (1) studies including patients with benign gynecological pathology; (2) studies including patients who underwent robotic single-site hysterectomy (RSS-H); (3) studies reporting at least one outcome of interest; (4) peer-reviewed articles, published originally.

Nonoriginal studies, preclinical trials, animal trials, abstract-only publications, and articles in languages other than English were excluded.

Concerning the main outcomes of interest, quantitative variables were median presurgical time, console time, operative time, blood loss, and hospital stay. The qualitative variables were the conversion rate to LPS/LPT, reoperation, and intraoperative and postoperative complications. Stratification of complications according to the Clavien–Dindo classification was also evaluated [13].

Among the studies that carried out a comparison of surgical techniques, we selected only the RSS-H data. The studies selected and all reasons for exclusion are mentioned in the preferred reporting items for systematic reviews and meta-analyses (PRISMA) flowchart (Figure 1). All included studies were assessed regarding potential conflicts of interest.

### 2.3. Quality Assessment

Assessment of the quality of the included studies was conducted using the Newcastle–Ottawa scale (NOS) [14]. This assessment scale uses three broad factors (selection, comparability, and exposure), with the scores ranging from 0 (lowest quality) to 8 (best quality). Two authors (G.R. and F.P.) independently rated the study’s quality. Any disagreement was subsequently resolved by discussion or consultation with C.R. The NOS scale is reported in Appendix A.

## 3. Results

### 3.1. Studies’ Characteristics

From database screening, 219 studies were selected. Of those, eight matched inclusion criteria and were included in our review. The basic characteristics of the included studies (first author and year of publication, country, study design, years range of study, and the number of participants) are described in Table 1. All the studies used the da Vinci system, with different models. The da Vinci Si surgical system was the surgical robotic system used in six of the included studies. J. Jayakumaran [15] used da Vinci Xi surgical system, the fourth-generation robot. Misal [16] used the SP1098 da Vinci SP surgical system.

### 3.2. Patients’ Characteristics

The characteristics of patients are summarized in the Table 2. A total of 212 patients were analyzed. Mean patients’ age was 45.42 years old (range 28–49.5 years old) and mean BMI was 25.74 (range 22–28.5 kg/m^2^). A total of 81 over the 212 patients had a history of previous abdominal surgery. The most frequent cause of hysterectomy was uterine benign neoplasm (83 patients) and adenomyosis (32 patients). 16 patients underwent surgery for pelvic pain and 10 patients for the treatment of female-to-male transsexualism. Four patients had endometrial hyperplasia, four had a genetic risk of cancer, four had uterine bleeding, two had uterine prolapse, two had CIN, one had cervical dysplasia, and at least one patient had a paratubal Mullerian cyst. The uterine pathology is not reported for 53 patients.

### 3.3. Outcomes

The main studies’ outcomes are presented in Table 3. The mean presurgical time, including port placement and docking time was 15.56 (range 3–30) min. Two studies (Lopez et al. [20] and Misal et al. [16]) did not report this variable. Mean console time was reported in 6 studies and is 83.21 min (range 25–180 min). The mean operative time was 136.6 min (range 60–294 min) and the mean blood loss was 43.68 mL (range 15–300 mL). Conversion to standard LPS or LPT did not occur in five studies. Two works (Jayakumaran et al. [15] and Lopez et al. [20]) reported, respectively, two and eight cases of conversion to robotic-assisted multiport surgery. Only two studies reported 1 case of intraoperative complications each (both studies, Jayakumaran et al. [15] and Lopez et al. [20], reported accidental cystotomy with a rate of 2.9% and 2%, respectively).

The overall postoperative complications occurred in three patients. Three studies (Lopez et al. [20], Jayakumaran et al. [16], and Chien-Wen et al. [22]) did not report this variable. Bagliolo et al. [17] reported a minor postoperative complication in one case: vaginal bleeding. Bagliolo et al. [21] described only one grade II complication (according to the Clavien–Dindo classification [13]). In another study (Misal et al. [16]), one patient reported vaginal bleeding in postoperative week three, which did not require any intervention. Median hospital stay is 1.71 days (range 0.96–3.5 days). No patient was reoperated and rehospitalized within the next 30 days.

### 3.4. Direct Comparison with Other Techniques

Lopez et al. [20] compared perioperative outcomes between laparoendoscopic single-site surgery (LESS) and robotic single-site surgery (RSSH) for benign indications, showing a statistical significance in an increase in total the operative time (121.0 ± 31.7 vs. 139.3 ± 45.8, *p* = 0.002) and a decrease in the length of the hospital stay (31.9 ± 14.8 vs. 23.3 ± 9.1, *p* = 0.003) for the RSSH group. In one of the included studies, Akdemir et al. [18] presented the early surgical outcomes in twenty-four patients undergoing robotic single-site total hysterectomy (RSS-TH) compared to thirty-four patients undergoing laparoendoscopic single-site total hysterectomy (LESS-TH). The study reported a significant difference in the mean total operation time among patients who had a laparoendoscopic and a robotic approach (86 vs. 98.5, *p* = 0.013). In contrast, vaginal closure time was higher in the LESS group compared to the RSS group (26.5 vs. 21, *p* = 0.011). The estimated blood loss, the postoperative pain score, and the length of hospitalization after surgery were similar in the two groups. Paek et al. [19] presented similar results, comparing LESS-H and RSS-H in treating benign gynecologic disease. The RSS-H group had a longer operation time (170.9 ± 65.5 vs. 88.3 ± 38.4, *p* < 0.0001), lower estimated blood loss (20 vs. 50 mL, *p* = 0.040), and no major or minor perioperative complications, compared to the LSS-H group (1.4% complication rate). Two studies, included in our review, compared single-site and multiport robotic surgery. Bogliolo et al. [21] matched 40 women undergoing single-site hysterectomy and bilateral salpingo-oophorectomy (BSO) (RSSH) with 59 women undergoing standard multiport hysterectomy and BSO (RH). The docking time was significantly lower in the RH group (9 ± 5 vs. 17 ± 6, *p* < 0.001), without a difference in the total operating time. The operative bleeding and the hospitalization stay were lower in the RSSH group (respectively: 46 ± 52 vs. 150 ± 151, *p* < 0.001 and 1.5 ± 1 vs. 2.5 ± 2, *p* = 0.009). In the same way, the postoperative pain at 24 h from surgery (T1) was significantly lower in the RSSH group (0 vs. 1, *p* = 0.009). Chen et al. [16] compared single-site and multiport robotic hysterectomy (24 RSSH vs. 57 RMSH) for benign gynecological conditions. In the RSSH group, the operative time was significantly shorter (140.3 ± 34.4 vs. 172 ± 52.7, *p* = 0.002) and the docking time was significantly longer (15.8 ± 5.5 vs. 8.6 ± 2.5, *p* < 0.001). While intraoperative blood loss and hemoglobin drop were similar, the length of hospitalization appeared lower in the RSSH group (2.4 ± 0.7 vs. 3.0 ± 0.9, *p* = 0.005). There was no intraoperative complication or readmission within 30 days in both groups.

## 4. Discussion

The relevance of the findings of our study is related to two main factors: improvement in the operative qualitative and quantitative variables on the one side and aesthetic outcomes on the other. These results highlight the role the RSSH technique could represent in gynecological practice for benign pathologies.

Total hysterectomy is a widespread procedure and minimally invasive approaches are becoming the preferred route [23,24]. The advantages are a less invasive procedure, leading to a reduction in intra and postoperative complications, such as fewer bleedings, fewer port related injuries, decrement in postoperative pain, and faster recovery [8]. In addition, the image of the pelvis is magnified and there is the possibility of using a 3D system of visualization that improves the surgical performance, allowing more precise and accurate movements. That is extremely supportive if we consider the anatomical structures that risk being fatally damaged during the procedure such as the ureters, the iliac arteries, the aorta, and the bladder, structures that the minimally invasive approach and visualization allow to better isolate and work on compared to an open surgery [25].

Reviews of the literature demonstrate how MIS is safe and effective and it is related to a superior quality of life in the postoperative period compared to the abdominal approach [26].

The single site surgery is an enhancement of the traditional multiport site surgery because it combines the advantages of the endoscopic technique with the possibility of using a single incision to access the abdominal cavity. For instance, using a single incision may relate to higher cosmetic outcomes and patients’ satisfaction [27,28]. The use of a single port site to access the abdominal cavity with several instruments has an impact on the quality of life, since the reduction in skin scars has a psychological impact, especially in patients suffering from diseases that lead to such a demolitive surgery. This aspect is relevant if we consider the mean age of the patients of our study undergoing hysterectomy for benign causes. Age range is between 28 and 49.5 years old, which is a relatively young age. A 2015 study focused on the perception of single site surgery on body image in two groups of women, under and over 40 years old. The study used the body image scale (BIS) to measure perception and satisfaction of women with their bodily appearance after surgery. The results showed that both groups preferred a surgery that takes advantage of the use of a natural orifice such as the umbilical scar, allowing a “scar free” surgery [29].

Robotic surgery is gaining ground among surgical techniques in gynecology. The first robotic single-site hysterectomy was described by Kane et al. in 2010 who successfully performed a hysterectomy in a young woman affected by endometriosis [30]. In RSS, movements are controlled by the main operator and carried out by the robot. Another point is that the learning process on the da Vinci surgical system is faster compared to laparoscopy, and this allows surgeons that are not adept at standard laparoscopy to approach minimally invasive procedures [11,31]. Furthermore, though further data are needed, complex surgeries and surgeries on obese patients seem to benefit from better performance thanks to robotics [31]. Even though RSS is still at its outset and its advantages compared to LESS are not clear yet, its costs are lower compared to multiport surgery and may be equivalent to LESS and its peri and post operative outcomes are proven to be comparable with the endoscopic approach [31,32,33].

## 5. Conclusions

The review of the scientific literature on this topic demonstrates how RSS is a safe and feasible option in performing total hysterectomy in patients affected by benign gynecological conditions. The use of RSS has the potential to associate acceptable complication rates with higher aesthetic outcomes compared to the multiport laparoendoscopic technique thanks to the use of a single port incision. Nevertheless, higher quality studies are warranted to demonstrate the pros and cons of robotic single site hysterectomy for benign pathologies and more data are needed to determine advantages and disadvantages compared to other routes.

In conclusion, results of this work stress the potential role RSS can have in the routine treatment of benign conditions that require hysterectomy among the other MIS approaches.

## Figures and Tables

**Figure 1 medicina-59-00411-f001:**
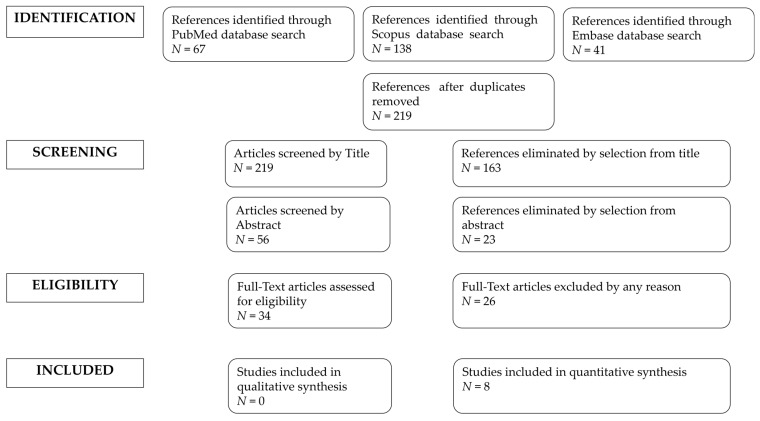
Preferred reporting items for systematic reviews and meta-analyses (PRISMA) flowchart.

**Table 1 medicina-59-00411-t001:** Studies’ characteristics.

Autor; Year	Country	Study Design	Years Range	*N* of Participant	Type of Surgery	Type of Robot
S. Bogliolo, 2014 [17]	Italy	Retrospective cohort study, monocentric	April 2013 to December 2013	10	Total hysterectomy + bilateral adnexectomy	da Vinci Si surgical system
A. Akdemir, 2015 [18]	Turkey	Retrospective case-control study, monocentric	January 2012 to December 2013	24	Total hysterectomy	da Vinci Si surgical system
J. Paek, 2015 [19]	Korea	Retrospective case-control study, monocentric	March 2011 to December 2014	25	Total hysterectomy	da Vinci Si surgical platform
S. Lopez, 2015 [20]	USA	Retrospective cohort study, multicentric	18 March 2013 to 30 December 2013	50	Total hysterectomy	da Vinci surgical system
S. Bogliolo, 2016 [21]	Italy	Retrospective case-control study, monocentric	March 2011 to October 2014	45	Total hysterectomy +/− bilateral adnexectomy	da Vinci Si surgical system
J. Jayakumaran, 2017 [15]	UK	Retrospective cohort study, monocentric	June 2016 and January 2017	24	Total hysterectomy +/− bilateral adnexectomy	da Vinci Xi robotic system
C. Chien-Wen, 2019 [22]	Taiwan	Retrospective case-control study, monocentric	June 2014 to December 2017	26	Subtotal hysterectomy	da Vinci Si surgical system
M. Misal, 2020 [16]	USA	Retrospective cohort study, monocentric	December 2019 to March 2020	8	Total or radical hysterectomy	da Vinci SP surgical system

**Table 2 medicina-59-00411-t002:** Clinical characteristics of patients.

Author; Year	Patient *n*	Median Age, yr (Range)	Median BMI kg/m^2^ (Range)	Previous Abdominal Surgery, *n* of pt (%)	Presence of Adhesion, *n* of pt (%)	Cause of Hysterectomy, *n* of pt (%)	Uterine Weight, gr (Range)
S. Bogliolo, 2014 [17]	10	28 ± 5.7 (20–40)	22 ± 1.7 (19–25)	4 (40)	NA	Female-to-male transsexualism 10 (100)	89 ± 15 (60–120)
A. Akdemir, 2015 [18]	24	49.5 (40–61)	28.5 (21.7–34.2)	18 (75)	NA	NA	192.5 (65–520)
J. Paek, 2015 [19]	25	48.0 ± 4.1 (NA)	24.3 ± 2.5 (NA)	16 (64)	12 (48)	Leiomyoma: 16 (64); Adenomyosis: 3 (12); CIN: 2 (8); Endometrial hyperplasia: 4 (16)	271 ± 119 (NA)
S. Lopez, 2015 [20]	50	46.0 ± 9.4 (NA)	25.9 ± 6.1 (NA)	NA	NA	NA	125.6 ± 68.5 (NA)
S. Bogliolo, 2016 [21]	45	46 ± 10.9 (34–64)	25 ± 5 (18–38)	38 (84.4)	NA	Uterine myomas adenomyosis genetic risk of cancer * 33 (73.3) 5 (11.1) 4 (8.9)	137 ± 39 (NA)
J. Jayakumaran, 2017 [15]	24	45 (17–70)	27 (18.4–41.9)	NA	1 (4.2%)	Pelvic pain 16 (45.7%) Fibroids 11 (31.4%) Adenomyosis 2 (5.7%) Uterine prolapse 2 (5.7%) Bleeding 2 (5.7%) Cervical dysplasia 1 (2.9%) Paratubal/mullerian cyst 1 (2.9%)	176 (46–532)
C. Chien-Wen, 2019 [22]	26	43.9 ± 5.9 (34–60)	23.0 ± 3.4 (18.0–32.4)	NA	10 (38.5)	Myoma: 20 (76.9) Adenomyosis: 20 (76.9)	264.6 ± 140.9 (100–750)
M. Misal, 2020 [16]	8	46.3 ± 13.6 (NA)	27.8 ± 7.5 (22.2–40)	5	NA	Leiomyoma: 1; leiomyoma + endometriosis: 1; leiomyoma + paratubal cyst: 1; adenomyosis: 1; adenomyosis + endometriosis: 2; postmenopausal bleeding:2	136.1 ± 61.5 (87–246)

Yr: year; BMI: Body Mass Index; pt: patients; gr: grams; NA: Not Available; CIN: Cervical Intraepitelial Neoplasia. * women with genetic risk for gynecological cancer and uterine pathology.

**Table 3 medicina-59-00411-t003:** Studies’ outcomes (intraoperative and postoperative).

Author; Year	Median Pre-Surgical Time *, min (Range)	Median Console Time, min (Range)	Median Operative Time, min (Range)	Median Estimated Blood Loss, mL (Range)	Intra-Operative Complications, *n* (%)	Postoperative Complications (%)	Conversion to Multiport/LPS/LPT	Median Hospital Stay, d ± SD (Range)	Reoperation (%)	Readmission within 30 Days, *n*
S. Bogliolo, 2014 [17]	9 ± 2 (6–18) §	79 ± 15 (55–110)	137 ± 32 (90–210)	30 ± 24 (15–100)	0	1 (0.1)	0	2.4 ± 0.9 (2–5)	NA	NA
A. Akdemir, 2015 [18]	13.5 (3–11)	74.5 (60–160)	98.5 (71–183)	22.5 (40–61)	0	0	0	1.6 (1–3)	NA	NA
J. Paek, 2015 [19]	14.0 ± 4.7	99.6 ± 49.7	170.9 ± 65.5	20 (30)	0	0	NA	3.5 ± 0.7	NA	NA
S. Lopez, 2015 [20]	NA	NA	139.3 ± 45.8	37.2 ± 30.7	1 (2)	NA	8 (16.0)	0.96 ± 0.3 (NA)	NA	NA
S. Bogliolo, 2016 [21]	17 ± 6 (7–30) §	115 ± 26 (NA)	144 ± 41 (82–265)	46 ± 52 (10–200)	0	1 (2.2%)	0	1.5 ± 1 (NA)	NA	NA
J. Jayakumaran, 2017 [15]	19 (4–47)	41 (25–120)	132 (60–294)	75 (20–300)	1	NA	2	0.96 (0.96–5)	NA	NA
C. Chien-Wen, 2019 [22]	15.8 ± 5.5 (9–28) §	61.1 ± 35.6 (25–180)	140.3 ± 34.4 (63–205)	71.2 ± 40.4 (50–200)	0	NA	0	2.4 ± 0.7 (1–4)	NA	0
M. Misal, 2020 [16]	NA	NA	86.5 ± 27.1 (60–132)	NA (20–100)	0	1	0	NA	0	0

* Median pre-surgical time includes port placement and docking time. § only docking time. SD: standard deviation; d: days; LPS: Laparoscopy; LPT: Laparotomy.

## Data Availability

Not applicable.

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
