# Peer review of "Robotic Single-Site Hysterectomy in Gynecologic Benign Pathology: A Systematic Review of the Literature"

_medicina, 2023, doi:10.3390/medicina59020411_

Round 1

Reviewer 1 Report

Dear Authors, thank you for giving me the opportunity to review your manuscript. The general impression made by this publication - it is perfunctory. It should be revised substantially. Please find some comments below:

1.     Please write before period the reference numbers in the text body.

2.     The flowchart is hard to read because of the poor quality of the figure.

3.     Tables are very hard to read. This form of graphic representation of the tables is unacceptable. 

4.     Please remove extensive spacing (for example line 97).

5.     Line 190-198 – What is the point of explaining the steps of the hysterectomy in the discussion?

6.     Line 218 – What do you mean by “demolition surgery”?

7.     English needs to be improved – there are words that are misused – for example – line 220 – “low age”? Did you mean young age? 

8.     Line 228-300 – You already mentioned that in the introduction.

9.     Line 237 – benefit from

10.  Line 251 – routine treatment

11.  The topic of this paper is very interesting and up to date, however, the manuscript in its current form is raw and it still needs a lot of work. The discussion is vague, it seems like the cosmetic effect is the only advantage of RSS. There are no major conclusions from that review. For this manuscript to be a good fit for the Journal, the whole discussion and conclusions section should be revised and changed.

Author Response

Dear Reviewer,
Thank You for taking the time to review our manuscript and for your comments. They are crucial
and valuable to us in raising the quality standard of our work.
We wanted to inform You that we have made a general revision of the English and grammar. In
addition, a specification for Your revisions is below. We followed your advice and changed it
according to your observation:
1. Please write before period the reference numbers in the text body.
Thank You for Your suggestion. We modified the mistake about the reference numbers
before the period.
2. The flowchart is hard to read because of the poor quality of the figure.
Thank You for Your suggestion. We used a new PowerPoint picture for the flowchart, with
a higher resolution of the image.
3. Tables are very hard to read. This form of graphic representation of the tables is
unacceptable. 
Thank You for Your suggestion. The tables’ graphic has been changed as You can find in
the revised manuscript
4. Please remove extensive spacing (for example line 97).
Thank You. Your suggestion made our work easier to be read. Extensive spacing in line 97
has been removed, and the format of the manuscript has been reviewed
5. Line 190-198 – What is the point of explaining the steps of the hysterectomy in the
discussion?
We agree that that sentence was not crucial in the discussion. We removed lines 190-198
6. Line 218 – What do you mean by “demolition surgery”?
Sorry for the mistake. We intended “Demolitive” Line 221 Demolitive surgery such as a
total hysterectomy is, compared to conservative treatment (for example myomectomy for
the treatment of uterus fibromatosus)
7. English needs to be improved – there are words that are misused – for example – line 220 –
“low age”? Did you mean young age? 
Thank You. We performed an extensive editing and check of the English grammar using an
internal service of our department
8. Line 228-300 – You already mentioned that in the introduction;
11. The topic of this paper is very interesting and up to date, however, the manuscript in its
current form is raw and it still needs a lot of work. The discussion is vague, it seems like the
cosmetic effect is the only advantage of RSS. There are no major conclusions from that

review. For this manuscript to be a good fit for the Journal, the whole discussion and
conclusions section should be revised and changed.
We rewrote the discussion and conclusion in order to avoid repetitions and to explain how
RSSS can be a valid method in performing a hysterectomy
9. Line 237 – benefit from;
10. Line 251 – routine treatment
We changed as suggested

Also, you can find the rewritten and corrected manuscript version in the attached file. We
highlighted any changes made.
Thank you very much for your advice and comments. We hope we have complied with your
requests.

Reviewer 2 Report

The review is well done with rigorous methods

I have just few suggestions :

INTRODUCTION

Line 47-48 ”Thanks to this procedure, it is possible to reduce the number of incisions and improve aesthetic outcomes.[7, 8]”.                                                          

Reducing the number of incisions there would  be also a reduction of port insertion morbidity as scar infections, haematoma, nerve and vessel lesions e not only aesthetic improvement.

I suggest following References: doi: 10.3390/jcm10102073.

                                                             doi: 10.1002/rcs.1613

REFERENCES: check reference N° 22

Author Response

Dear Reviewer,
Thank You for taking the time to review our manuscript and for your comments. They are crucial and valuable to us in raising the quality standard of our work.
We wanted to inform You that we have made a general revision of the English and grammar. In addition, We followed your advice and changed it: Line 47-48 ”Thanks to this procedure, it is possible to reduce the number of incisions and improve aesthetic outcomes.[7, 8]”. Reducing the number of incisions there would  be also a reduction of port insertion morbidity as scar infections,
hematoma, nerve and vessel lesions e not only aesthetic improvement. Moreover, we read the references suggested, and we found them pertinent to the topic of the review, for this reason, we are pleased to introduce them in our work (references 9 and 16). You can find the rewritten and corrected version of the manuscript in the attached file. We highlighted any changes made.
Thank you very much for your advice and comments. We hope we have complied with your requests.
